# Discovery of *epi*-Enprioline as a Novel Drug for the Treatment of Vincristine Resistant Neuroblastoma

**DOI:** 10.3390/ijms21186577

**Published:** 2020-09-08

**Authors:** Wondossen Sime, Mohamed Jemaà, Yasmin Abassi, Vito Alessandro Lasorsa, Julie Bonne Køhler, Karin Hansson, Daniel Bexell, Martin Michaelis, Jindrich Cinatl, Daniel Strand, Mario Capasso, Ramin Massoumi

**Affiliations:** 1Department of Laboratory Medicine, Translational Cancer Research, Lund University, Medicon Village, 223 81 Lund, Sweden; wondossen.sime@med.lu.se (W.S.); jemaamohamed@gmail.com (M.J.); yasmin.abassi@tron-mainz.de (Y.A.); julie.bonne_kohler@med.lu.se (J.B.K.); karin.hansson.3064@med.lu.se (K.H.); daniel.bexell@med.lu.se (D.B.); 2Dipartimento di Medicina Molecolare e Biotecnologie Mediche, Università degli Studi di Napoli Federico II, Via Sergio Pansini 5, 80131 Naples, Italy; lasorsa.alessandro@gmail.com (V.A.L.); mario.capasso@unina.it (M.C.); 3CEINGE Biotecnologie Avanzate, Via G Salvatore, 80131 Naples, Italy; 4School of Biosciences, University of Kent, Canterbury CT2 7NJ, UK; M.Michaelis@kent.ac.uk; 5Institute of Medical Virology, Clinics of the Goethe-University, D-60596 Frankfurt am Main, Germany; cinatl@em.uni-frankfurt.de; 6Centre for Analysis and Synthesis, Department of Chemistry, Lund University, 221 00 Lund, Sweden; daniel.strand@chem.lu.se; 7IRCCS SDN, Via Emanuele Gianturco, 113, 80143 Naples, Italy

**Keywords:** chemoresistance, neuroblastoma, *epi*-enprioline, apoptosis, vincristine

## Abstract

Neuroblastoma is a childhood solid tumour originating from undifferentiated neural progenitor cells of the sympathetic nervous system. Drug resistance of childhood cancer neuroblastoma is a serious clinical problem. In the present study, we aimed to identify novel drugs that can inhibit the growth and survival of chemoresistant neuroblastoma. High-throughput screening identified a small molecule, *epi*-enprioline that was able to induce apoptosis of vincristine-resistant neuroblastoma cells via the mitochondrial apoptotic pathway. *Epi*-enprioline reduced tumour growth in multiple preclinical models, including an orthotopic neuroblastoma patient-derived xenograft model in vivo. In summary, our data suggest that *epi*-enprioline can be considered as a lead compound for the treatment of vincristine-resistant neuroblastoma uncovering a novel strategy, which can be further explored as a treatment for drug-resistant neuroblastoma.

## 1. Introduction

Treatment strategies for neuroblastoma (NB) patients include chemotherapy, including cyclophosphamide, cisplatin, doxorubicin, etoposide, carboplatin, and vincristine, either in combination or alone followed by surgical resection, myeloablation, and autologous stem cell rescue, radiation, and immunotherapy [1,2]. Recurrence in high-risk patients is observed in more than 50% of children with high-risk disease [3]. Since options for tackling chemoresistant NB in clinical practice are limited, there is an urgent need for developing additional treatment strategies.

Treatment failure due to multi-drug resistance against chemotherapeutic agents is seen for over 90% of metastatic cancer patients [4,5]. Resistance to a given drug can be either pre-existing (intrinsic resistance) [6] or occur during the course of treatment (acquired resistance) [7]. There are numerous mechanisms involved in the development of multiple drug resistance [7,8]. These can be summarised as increased removal of drugs by transporters, enhanced drug metabolism, reduced cellular uptake, changes in the expression of drug targets, intracellular drug sequestration, and changes in the expression of genes involved in cell death, cell cycle, or DNA repair. Other processes that have been linked to drug resistance include mitochondrial alteration, autophagy, epithelial-mesenchymal transition, and cancer cell stemness. Among these mechanisms, increased expression of efflux pumps, which leads to decreased intracellular drug concentrations, is the most studied [9,10].

Drug resistance due to alteration in cell cycle checkpoints is mostly seen in combination therapies [11]. Reduced intracellular activation of pro-drugs and drug detoxification by the cytochrome P450 enzymes and glutathione S-transferases, respectively, are other well-studied mechanisms of chemoresistance [12,13]. The altered expression of pro-apoptotic proteins (BAX and BAK) or anti-apoptotic proteins (BCL2, BCL-XL and MCL1) can directly affect drug resistance of cancer cells [14,15]. The epithelial-mesenchymal transition process characterised by a dissociation of cell–cell contacts and alteration of the cytoskeletal network was recently shown to mediate drug resistance in different types of cancer cells [16,17].

In the present study, we have used previously established vincristine-resistant NB cell lines to screen for novel drugs against treatment-resistant NB. Our high-throughput screening identified *epi*-enprioline, which could limit the growth of vincristine-resistant NB by inducing cell death via the intrinsic mitochondrial pathway of apoptosis.

## 2. Results

To identify a new anti-cancer drug that can affect the survival of vincristine-resistant neuroblastoma (NB) cells (VCR-20), we performed cell survival screening assay by using 10 µM of each 1360 small molecules from the National Cancer Institute (NCI). Eleven molecules could reduce the survival of resistant cells by 50% or more (Figure 1A,B). These identified small molecules were subsequently tested in a concentration-dependent cell survival assay. While no differences in cell survival could be detected when cells were grown at 0.01 µM of the small molecules, a significant difference could be observed after treatment with 0.1 µM of the racemic *epi*-enprioline (Figure 1C). To gain insight into the structural features of *epi*-enpiroline, a single crystal X-ray diffraction (scXRD) structure was solved and compared to the known structure of enpiroline (Figure 1D and Appendix A). The electrostatic potential surface map of *epi*-enpiroline was also calculated to probe the binding motif of this structure (Figure 1E).

Treatment of vincristine-resistant NB cells with *epi*-enprioline reduces both proliferation and number of surviving cells (Figure 2A,B), as well as the number of colonies (Figure 2C). The sensitivity of other vincristine-resistant NB cells to *epi*-enprioline was confirmed using SK-N-AS-VCR-20 (IC_50_ = 1.2 ± 0.7 µM) and UKF-NB-4-VCR-50 (IC_50_ = 3.05 ± 0.7 µM). Treatment of these chemoresistant NB cells with *epi*-enprioline reduced proliferation and cell survival (Figure 2D–G). Furthermore, LU-NB-2 PDX cells, derived from a NB brain metastasis following treatment relapse were highly sensitive for *epi*-enprioline with an IC_50_ value of 0.5 µM (Figure 2H). Treatment with *epi*-enprioline caused markedly decreased LU-NB-2 PDX cell viability (Figure 2I). In addition, *epi*-enprioline was not affecting the survival of primary embryonic fibroblasts (Appendix A).

The early effect of *epi*-enprioline (after 24 h) on cell death and cell cycle was demonstrated for VCR-20 cells, where the frequency of cells in subG1 was higher in 0.5 µM *epi-*enprioline-treated compared to control cells (Figure 3A). This effect was even more pronounced after treatment with 1.0 µM of *epi*-enprioline (Figure 3A). Flow cytometry analysis showed *epi*-enprioline dose-dependent cell shrinkage characterised by a decrease of cell size and gain of granularity (Figure 3B), which is a hallmark of cells undergoing apoptosis [18]. Annexin V and PI co-staining followed by flow cytometry analysis also demonstrated *epi-*enprioline-induced apoptosis (Figure 3C). Changes in intracellular calcium are associated with apoptosis via mitochondrial membrane disruption [19]. We used the calcium-sensitive fluorescence probe Fluo-3/AM to monitor changes in intracellular calcium levels. The fluorescence emitted from *epi*-enprioline-treated cells was shifted to a higher intensity compared to control cells, indicating an increase in the intracellular calcium concentration (Figure 3D). Furthermore, we used the potential-independent mitochondria-specific vital dye MitoTracker Green to measure the accumulation of depolarised mitochondria [20]. MitoTracker Green is a lipophilic thiol-reactive dye that selectively labels the mitochondrial inner membrane and matrix for measuring of cellular mitochondrial content. The increase of MitoTracker Green fluorescence is a sign of mitochondrial depolarisation due to a loss of mitochondria potential. Analysis by flow cytometry revealed a threefold and sevenfold increase in MitoTracker Green staining of cells treated with 0.5 and 1.0 µM *epi*-enprioline, respectively (Figure 3E). Further, DiOC_6_ staining followed by flow cytometry analysis confirmed the loss of mitochondrial membrane potential (ΔΨm) in *epi*-enprioline-treated compared to control cells (Figure 3F). Activation of caspase 3 in *epi*-enprioline-treated cells demonstrated the execution-phase of cell apoptosis (Figure 3G). These results suggest that *epi*-enprioline interferes with the intrinsic mitochondria-dependent apoptosis pathway in vincristine-resistant NB cells.

Next, we subcutaneously implanted VCR-20 cells in nude mice. Caliper measured visible tumours were treated with *epi*-enprioline or DMSO control. This treatment resulted in reduced volume of *epi*-enprioline-treated tumours compared to control mice (Figure 4A). This proof-of-concept experiment suggest that *epi*-enprioline has potential as lead candidate. As expected, an elevated number of cleaved caspase 3 immunoreactive cells was observed in tumour cells isolated from *epi*-enprioline-treated mice compared to control animals (Figure 4B). In another set of experiments, VCR-20 was injected orthotopically into the adrenal gland of mice before intraperitoneal (i.p.) treatment with *epi*-enprioline. In vivo imaging using IVIS-CT scan demonstrated that *epi*-enprioline-treated mice show reduced tumor growth compared to control mice (Figure 4C,D) and prevented tumor metastasis to distant organs (Figure 4E) without any significant changes in the weight of the animals (Appendix AA). Finally, LU-NB-2 PDX cells were injected orthotopically into the adrenal gland and mice were treated i.p. with *epi*-enprioline. Treated mice showed decreased tumor growth as compared to controls (Figure 4F,G), without any significant changes in the weight of the animals (Appendix AB). Importantly, tumor metastasis to distant organs could only be observed in control mice (Figure 4H; Videos S1,2).

To examine whether *epi*-enprioline can change the levels of vincristine resistance-associated genes, we performed expression profiling of resistance-associated genes following *epi*-enprioline treatment. By analysing gene expression data we set the cutoff for the fold change at +/−2.0 and evaluated the following contrasts: (i) VCR-10 vs. parental (26 genes); (ii) VCR-10-enprioline vs. parental (29 genes); and (iii) VCR-10-enprioline vs. VCR-10 (five genes). Differentially expressed gene lists were then compared to identify commonly and uniquely regulated genes (Figure 5). VCR-10 vs. parental contrast showed only five unique genes (all upregulated). Among the eight unique genes in VCR-10-enprioline vs. parental, we found members of ABC-transporters and cytochrome enzymes. Of note, *CYP1A1* was the unique downregulated gene in all three of the contrasts we compared. We next performed GO and KEGG Pathway Enrichment Analysis with gene lists from the above contrasts. Appendix A reports the top ten of enriched GO terms and pathways (full results in Appendix A). In VCR-10 vs. parental and VCR-10-enprioline vs. parental, pathways in cancer was the most enriched KEGG pathway; in addition, several cancer types were found enriched in KEGG pathway analysis (Appendix A). GO molecular functions were overrepresented in terms related to protein kinase activities (Appendix A). Among the biological processes, we found a higher presence of apoptosis and cell death related GO terms in VCR-10-enprioline vs. parental contrast compared to VCR-10 vs. parental. By using the VCR-10-enprioline vs. VCR-10 list (five genes), we found the enrichment of xenobiotics and drug metabolic processes (GO molecular function) and chemical carcinogenesis (KEGG).

## 3. Discussion

Cure of children with high-risk NB disease is limited despite aggressive multimodal treatment strategies including surgery, radiotherapy, stem cell transplantation and chemotherapy. Similar to other cancer diseases, the acquisition of drug resistance by tumour cells is of great importance to developing novel drugs [3]. As MYCN amplification is indeed associated with poor prognosis in NB, we decided to use previously generated VCR cells, which is a MYCN amplified aggressive and metastasis cell line originating from NB patient. In addition to p53 mutations, these cells harbor other important pathogenic genomic aberrations such as 1p21.1–36.3 deletion and 9p21–24.3 as well as 3p22.1–25.3 loss. Vincristine is a vinca alkaloid that blocks cell growth by interfering with mitosis. Vincristine is included in the rapid COJEC (Cisplatin (C), vincristine (O), carboplatin (J), etoposide (E), and cyclophosphamide (C)) treatment regimen. In the present study, we investigated the influence of vincristine resistance on cell growth and to identify a novel inhibitor that can limit the growth of these cells.

Searching for an alternative anti-cancer drug that can affect the survival of VCR NB cells, we screened a library of 1360 small molecules. The screening identified 11 compounds with significant activity and among them, racemic *epi*-enprioline showed the highest cytotoxic effect. Structurally, *epi*-enpiroline is a diasteromer of enpiroline, a compound that was originally developed as an anti-malarial drug agent against multidrug resistance cases. Interestingly, both enantiomers of enpiroline exhibit antimalarial activity as they share binding motif with the natural products quinine and quinidine respectively [21,22]. The solid-state structure of *epi*-enpiroline was found to closely resemble that of enprioline with respect to conformation but the different stereochemistry of the two compounds reflects in a quite different binding motif. 

Since treatment of VCR NB cells with *epi-*enprioline resulted in growth retardation and reduced colony growth, we analysed the mechanism of cell survival. Increase of the subG_1_ fraction of the cell cycle, cell shrinkage, phosphatidylserine outer membrane translocation, as well as loss of mitochondrial membrane integrity, provided evidence of a pro-apoptotic function of *epi*-enprioline. Generally, apoptosis is divided into the extrinsic and intrinsic pathways. While the extrinsic pathway is mediated via cell surface death receptors, the intrinsic pathway is mediated through the mitochondria [23,24,25]. Applying different strategies, including measuring changes in intracellular calcium concentration (which is associated with apoptosis via mitochondrial membrane disruption), labelling of the mitochondrial inner membrane for measuring cellular mitochondrial content with MitoTracker and the measurement of the mitochondrial membrane potential (ΔΨm) using the DiOC_6_(3) dye, provided evidence that *epi*-enprioline interferes with the intrinsic mitochondria-dependent apoptotic pathway.

Gene expression profiling identified multiple genes up- or downregulated in *epi*-enprioline-treated VCR-10 cells in comparison to either parental or untreated VCR-10 cells. Overall, we observed downregulation of various genes in *epi*-enprioline-treated cells, including ABC-transporters such as *ABCB1* and *ABCG2*, which are known to be involved in drug resistance because of their role in the efflux mechanisms. Moreover, the cytochrome C P450 enzymes associated with drug metabolism, including CYP1A1, CYP3A4, CYP1A2, CYP2B6, CYP2C19, CYP2C8, anti-apoptotic gene *BCL2,* and other genes involved in cell cycle progression, such as *CDK2*, *CDK2D,* and *GSK3A*, were also downregulated in *epi*-enprioline treated Be2c-VCR cells. On the other hand, a number of genes such as *AHR*, *ESR1*, *CYP2E1*, *CYP2C9* and *NAT2* were shown to be upregulated in *epi*-enprioline-treated VCR-10 cells. GO enrichment analysis supported the findings reported above. Differentially expressed genes in *epi*-enprioline-treated VCR-10 cells resulted in terms related to apoptosis and response to xenobiotics among biological processes and protein kinases and drug binding molecular functions. Consistent with the regulation of several cytochrome C P450 enzymes in VCR-10-enprioline, we found the enrichment of the biological process; intrinsic apoptotic signalling pathway in response to DNA damage. Several cancer types were enriched among the KEGG pathways. A direct comparison between VCR-10-enprioline and VCR-10 showed the overrepresentation of GO categories involved in drug and xenobiotic metabolic process and response to xenobiotic stimulus.

To establish the anti-cancer effect and toxicity of *epi-*enprioline in vivo, we applied a xenograft transplantation experiment in athymic nude mice. *Epi*-enprioline treatment of mice bearing xenografts of the vincristine-resistant cell line resulted in significant decrease in relative tumour volume compared to mice treated with vehicle control. Furthermore, orthotopic transplantation of vincristine-resistant or PDX derived NB into the adrenal gland before treatment with *epi*-enprioline reduced tumor growth and tumor metastasis. These data suggest that *epi*-enprioline can be considered as a lead compound for a novel class of anti-cancer agents for the treatment of vincristine-resistant NB. Further studies need to establish whether the anti-cancer function of *epi*-enprioline is restricted to NB or whether it also applies to other chemo reagents and chemoresistant human cancer cells, in addition to future drug toxicology studies. To summarize, we identified a small molecule, *epi*-enprioline, that exerted cytotoxic effects against vincristine -resistant NB in vitro and in vivo.

## 4. Materials and Methods

### 4.1. Cell Culture

The human NB cell line SK-N-BE(2)-C (ATCC, CRL-2268, parental) was cultured at 37 °C and 5% CO_2_ in minimum essential medium (MEM, HyClone, Thermo Fisher Scientific, MA, USA) supplemented with 10% fetal bovine serum (FBS) (Sigma-Aldrich Sweden AB, Stockholm, Sweden) and 1% penicillin/streptomycin (GIBCO, MA, USA). To generate vincristine-resistant cells (Be2c-VCR), SK-N-BE(2)-C cells were cultured for 7 months with gradually increased concentrations of vincristine ranging from 1–10 ng/mL. The 10 ng/mL Be2c-VCR cells (VCR-10) were treated further with escalating concentration of 20 ng/mL (VCR-20) of vincristine for another month.

The non-*MYCN*-amplified neuroblastoma cell line SK-N-AS was kindly provided by Dr. Angelika Eggert (Universität Duisburg-Essen, Germany). The *MYCN*-amplified, *TP53*-mutant (C175F) neuroblastoma cell line UKF-NB-4 was derived from bone marrow metastases of a patient harboring a stage IV neuroblastoma [26,27]. The vincristine-resistant sub-lines were established by continuous exposure to step-wise increasing drug concentrations as previously described [28] and derived from the Resistant Cancer Cell Line (RCCL) collection (https://www.kent.ac.uk/stms/cmp/RCCL/RCCLabout.html). UKF-NB-4 cells were adapted to growth in the presence of vincristine 50 ng/mL (UKF-NB-4-VCR-50), SK-N-AS to vincristine 20 ng/mL (SK-N-AS-VCR-20). Cells were grown at 37  °C in Iscove’s modified Dulbecco’s medium supplemented with 10% heat-inactivated foetal calf serum and containing 100 IU/mL of penicillin and 100 μg/mL streptomycin. The vincristine-adapted sub-lines were continuously cultivated in the presence of the indicated drug concentrations.

LU-NB-2 patient-derived xenograft (PDX)-cells were established and characterized as previously described [29,30]. Briefly, PDX cells were derived from a NB brain metastasis following treatment relapse. Tumor cells contained 1p del, *MYCN* amplification and 17q gain. PDX cells were cultured as free-floating 3D-spheres in serum-free medium consisting of Dulbecco’s Modified Eagle’s Medium (DMEM) and GlutaMAXTM F-12 (3:1 ratio) supplemented with 1% penicillin/streptomycin, 2% B27 *w/o* vitamin A, 40 ng/µL basic fibroblast growth factor and 20 ng/µL epidermal growth factor.

Mouse embryonic fibroblast (MEF) cells were cultured in DMEM that was supplemented with 4500 mg/L l-glucose, 4 mm/L l-glutamine, and sodium pyruvate (hyclone, Thermo Scientific, Waltham, MA, USA), 20% fetal bovine serum (FBS), and 100 IU/mL penicillin and streptomycin mixture (Life Technologies, Carlsbad, CA, USA).

### 4.2. Drug Screening

VCR-20 cells were seeded in 96-well plates with a density of 5000 cells/well. Twenty-four hours later, cells were treated in triplicate with 10 µM of 1360 small molecules/well obtained from the National Cancer Institute (NCI) or dimethyl sulfoxide (DMSO) for 48 h. Cell viability (mean absorbance from three wells) was measured using the 3-(4,5-dimethylthiazol-2-yl)-2,5-diphenyltetrazolium bromide) tetrazolium reduction (MTT) assay and calculated as a fold change compared to DMSO-treated cells.

### 4.3. Animal Model

VCR-20 cells (2.0 × 10^6^ cells in 0.1 mL phosphate buffered saline (PBS)) were subcutaneously injected into the right flank of 4-week-old female NMR1-Foxb1 nude mice (Janvier Labs, Le Genest-Saint-Isle, France). Mice were randomly assigned into two weight-matched groups with 6 mice in each group receiving either *epi*-enprioline or DMSO. When the mice developed visible subcutaneous tumours, they were treated with *epi*-enprioline (10 mg/kg) every day for 12 days. Tumour growth was followed by measuring the volume every second day. The subcutaneous tumours were harvested, weighed and processed for further study. The experiment was terminated when the tumors reached a volume of 600–700 mm^3^. The animals were maintained under specific pathogen-free (SPF) conditions. All experimental procedures were approved by the Malmö and Lund Animal Ethics Committee with the ethical number M129-15 (Approval Date: 19 August 2015).

### 4.4. Establishment of Orthotopic Tumor and In Vivo Bioluminescence Imaging

Six to eight-weeks-old athymic nude mice were subjected to orthotopic implantation into adrenal gland with either VCR-20 GFP-Luc cells (1 × 10^6^) or with LU-NB-2 GFP-Luc PDX cells (2 × 10^6^). One week later, the tumor growth was monitored by non-invasive 2D bioluminescence (BLI) imaging, using IVIS-CT spectrum (PerkinElmer, MA, USA). Mice showing tumor growth signals were randomized into two groups (DMSO and *epi*-enprioline) based on their average BLI signal intensity recorded in a defined region of interest (ROI) with average total flux (photons/s) value of 7 × 10^5^ and 4 × 10^5^ among VCR-20 and LU-NB-2 implanted mice respectively. DMSO or *epi*-enprioline (10 mg/kg)-treatment was given intraperitoneally every three days up to the end points. Tumor growth and metastasis development was monitored using bioluminescence 2D and 3D μCT imaging. Briefly, mice anesthetized with 3% isoflurane gas injected intraperitoneally with 150 mg D-Luciferin/kg of body weight in PBS prior to imaging. Acquisition of 2D images was performed sequentially with a five-minute interval between different segments of exposures (emission: open filter, f/stop: 1, binning: 8). BLI signal intensity was quantified in Total flux (Photons/s) after deducting the average background signal (Bkg) from measurement region of interest (ROI) using the Living image analysis software (PerkinElmer, MA, USA). Ventral side 3D μCT images were taken to show distant metastasis other than the tumor growth within the implanting regions in the adrenal gland. The mouse organ atlas was overlaid on the 3D skeletal images together with images of mouse body slices including coronal, sagittal and transaxial view to show the possible metastatic anatomical sites using Living image analysis software (PerkinElmer, MA, USA).

### 4.5. Immunohistochemistry Staining

Formalin-fixed, paraffin-embedded xenograft tumor sections (5 µm) were deparaffinized using routine techniques, and placed in 200 mL of EnVisionTM target retrieval solution (pH 6.0; Dako, Hamburg Germany) for 20 min at 100 °C. After cooling for 20 min, slides were quenched with 3% H_2_O_2_ for 5 min before incubating with a primary antibody against cleaved caspase 3 (Abcam, ab2302, 1:500, Cambridge, Storbritannien) to detect apoptotic cells using a Dako Autostainer. Immunostaining was visualized using the EnVisonTM+ kit (Agilent Dako, Santa Clara, CA, USA). In addition, slides were also stained with hematoxylin and eosin.

### 4.6. RT^2^ Profiler PCR Array Analysis

Gene expression profiling related to human cancer drug resistance was performed using a 384-well RT^2^ Profiler PCR Array (PAHS-004Z, QIAGEN AB, Sollentuna, Sweden). Following total RNA extraction, cDNA was synthesised from Be2c-parental, VCR-10 and *epi*-enprioline (1 µM, 24 h)-treated VCR-10 cells based on the manufacturer’s instructions. Eighty-four different genes involved in cancer cell drug resistance were analysed based on SYBR Green real-time PCR using the QuantStudio™ 7 Flex (Applied Biosystems, CA, USA). Normalisation was performed using the five different housekeeping genes included in the array and the fold change was calculated using the RT^2^ Profiler PCR Array Data Analysis from Qiagen (http://www.qiagen.com/geneglobe).

### 4.7. Gene Ontology and Pathway Enrichment

Differentially expressed gene lists were used to perform Gene Ontology (GO) and Pathway Enrichment Analysis by using the tool WebGestalt (www.webgestalt.org). We used the overrepresentation enrichment analysis method to query the GO and the KEGG databases with the default options. *p*-values were calculated with the hypergeometric test and statistical significance level was set at false discovery rate (FDR) ≤ 0.05.

### 4.8. Proliferation and Viability Assay

Cells were seeded in 96-well plates with a 5000 cells/well density. After the treatment, 10 μL of alamarBlue reagent (ThermoFisher, MA, USA) was added to each well, mixed and then incubated for 4 h at 37 °C in a CO_2_ incubator. The absorbance of each sample was measured using a scanning microplate spectrophotometer reader (Synergy 2, Biotek, Winooski, VT, USA) by absorbance detection at 570 nm or fluorescence detection at excitation and emission wavelengths of 540–570 and 580–610 nm, respectively. IC50 values for each treatment were calculated using the GraphPad software. For cristal violet proliferation assay, cells were seeded in 96-well plates with a 5000 cells/well density. Twenty-four hours later, cells were treated or not with 0.5 µM epi-enprioline or DMSO for up to 72 h. Cells were washed once with PBS and fixed with 4% paraformaldehyde (PFA) for 15 min. The PFA was removed, and cells were stained for 30 min at room temperature with an aqueous solution containing 0.1% (*w*/*v*) crystal violet. Cells were washed three times with distilled water before the administration of 200 µL/well of 10% acetic acid and shaking with micropipettes. The absorbance of each sample was measured using a scanning microplate spectrophotometer reader (Synergy 2, Biotek) by absorbance detection at 595 nm.

### 4.9. Clonogenic Assays

To evaluate clonogenic survival, cells were seeded at different concentrations (from 100 to 2000 cells/well) in 6-well plates. Upon 24 h of adaptation time, cells were treated with different concentrations of epi-enprioline for 7 days in standard culture conditions. Colonies were then fixed with 4% PFA, stained with an aqueous solution of 1% (*w*/*v*) crystal violet and counted. Only colonies made up of >30 cells were included in the quantification. To evaluate clonogenic potential, cells were seeded at low concentrations (100 cells/well) in 6-well plates and cultured in standard conditions for 15 days. Colonies were then fixed with 4% PFA, stained with an aqueous solution of 1% (*w*/*v*) crystal violet and counted. Colonies made of >30 cells were included in the quantification, and colony size was evaluated using the ImageJ software.

### 4.10. Apoptosis and DNA Fragmentation Assay

For apoptosis analyses, cells were fixed in PFA on coverslips and stained with a Vindelöv solution containing propidium iodide (PI). After washing, the coverslips were mounted onto glass slides and cells were examined by fluorescence microscopy. Cells were scored for apoptosis based on nuclear morphology. Apoptosis was further evaluated using NucleoCounter NC-3000 (Chemometec, Allerod, Denmark) in conformity with the DNA fragmentation assay. Cells grown in 6-well plates were harvested by trypsinisation and pooled with the cells floating in the medium. After a short centrifugation, the supernatant was removed, and precipitated cells were washed once with PBS. After a second centrifugation, cells were resuspended in a small volume of PBS and the single-cell suspensions were added to 70% ethanol for fixation. The samples were vortexed and stored for 12–24 h at −20 °C. The ethanol-suspended cells were centrifuged and the ethanol carefully decanted. Cells were washed once with PBS and then resuspended in NucleoCounter Solution 3 (1 µg/mL DAPI, 0.1% Triton X-100 in PBS) followed by incubation for 5 min at 37 °C. Ten microliters of samples were loaded into a slide chamber (NC-slide A8) and the DNA fragmentation protocol was employed according to the manufacturer’s instructions (Chemometec, Allerod, Denmark).

### 4.11. Cytofluorometric Studies

Cytofluorometric acquisitions were performed by means of a FACSVerse cytofluorometer (BD Biosciences, CA, USA). Data were statistically evaluated using the Kaluza (Beckman Coulter, Indianapolis, IN, USA) software. Only events characterised by normal forward scatter (FSC) and side scatter (SSC) parameters were gated for inclusion in the statistical analysis after exclusion of cell doublets.

Cell cycle analysis: For the quantification of cell cycle profiling (DNA content), live cells were harvested, collected with the culture medium and resuspended in 0.3 mL pre-warmed growth medium supplemented with 2 mM Hoechst 33342 (Sigma-Aldrich) for 30 min at 37 °C in a CO_2_ incubator. Cell suspensions were analysed on a cytometer with ultraviolet excitation and emission collected at >450 nm. To quantify the apoptotic and DNA fragmented fraction, we measured the subG1 population of cells.Measurement of cell shrinkage: For the quantification of cell shrinkage, cells were harvested and collected with the culture medium before FACS assessment without any staining. Cell size was measured using the FSC and cell granularity using the SSC. Apoptotic cells are more granulated and smaller than live cells.Measurement of cell permeabilisatio: For the quantification of plasma membrane integrity, cells were harvested and collected with the culture medium and stained with 0.5 to 1 μg/mL 7-aminoactinomycin (7AAD, which only incorporates into dead cells, from ThermoFisher) for 30 min at 37 °C before FACS assessment.Measurement of cell scrambling and phosphatidylserine exposure: For the simultaneous quantification of plasma membrane integrity and phosphatidylserine exposure, cells were harvested and collected with the culture medium and stained with Annexin-V-FITC (1:200 dilution; ImmunoTools, Friesoythe, Germany) and 1 μg/mL PI (which only incorporates into dead cells, from Sigma-Aldrich) for 30 min at 37 °C before FACS assessment.Measurement of mitochondria accumulation: For staining mitochondria, cells were harvested and collected with the culture medium and labelled for 45 min at 37 °C with 100 nM of MitoTracker Green MTG (ThermoFisher, MA, USA) before FACS assessment. The signal shift is measured comparatively to non-treated cells.Measurement of intracellular calcium concentration: For the evaluation of cytosolic calcium, cells were collected and suspended in growth medium supplemented with 5 μM of the calcium tracker Fluo-3/AM (Biotium, Hayward, CA, USA). Cells were incubated at 37 °C for 30 min before calcium-dependent fluorescence intensity measurement. The Fluo-3/AM is measured with an excitation wavelength of 488 nm (blue laser) and an emission wavelength of 530 nm. The signal shift and the geometric mean were measured comparatively to non-treated cells.Measurement of cleaved caspase 3: To measure cleaved caspase 3, cells were fixed with 75% (*v*/*v*) ethanol in PBS, permeabilized with 0.25% (*v*/*v*) Tween 20 in PBS and stained with a FITC-conjugated Casp3 c (rabbit polyclonal #559341, BD-Bioscience).

### 4.12. Statistical Analyses

Statistical analyses were performed using the SigmaPlot or GraphPad Software. Results are expressed as mean ± SEM or as a percent. *p*-values *p* < 0.05*, *p* < 0.01**, and *p* < 0.001*** were deemed statistically significant. Statistical comparisons were assessed by analysis of variance (ANOVA) or by the Student’s *t*-test.

## Figures and Tables

**Figure 1 ijms-21-06577-f001:**
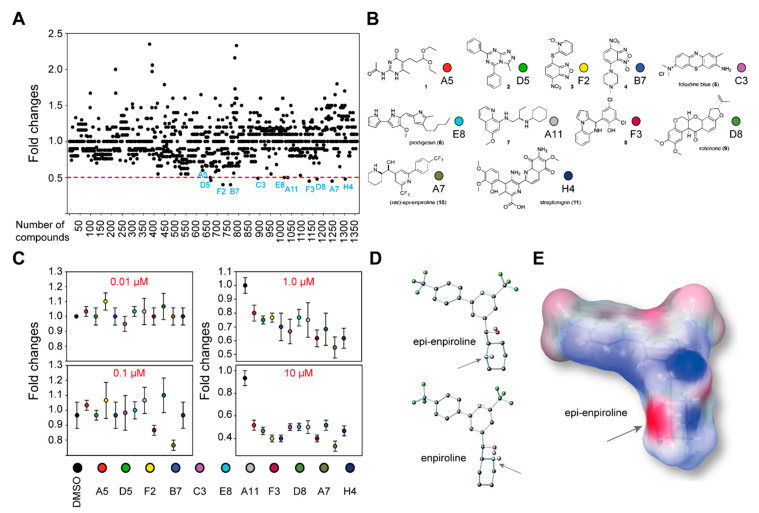
Identification of *epi*-enprioline as a selective inhibitor of neuroblastoma Be2c cells. (**A**) Be2c-VCR cells (VCR-20) were treated with small molecules or DMSO for 48 h. Cell viability was measured using the MTT assay and calculated as fold changes compared to DMSO-treated cells (*n* = two independent experiments in triplicate). (**B**) The chemical structure of molecules that could reduce the survival of resistant cells by at least 50%. (**C**) VCR-20cells were treated with different concentrations of molecules identified in Figure 1A. Cell viability was measured using MTT assay and calculated as fold changes compared to DMSO-treated cells (*n* = two independent experiments in triplicate, mean ± SEM). (**D**) Chemical structures of *epi*-enpiroline and enpiroline. Arrows highlight the piperidine nitrogen atom in each structure. (**E**) Electrostatic potential surface map of epi-enpiroline in a low-energy conformation.

**Figure 2 ijms-21-06577-f002:**
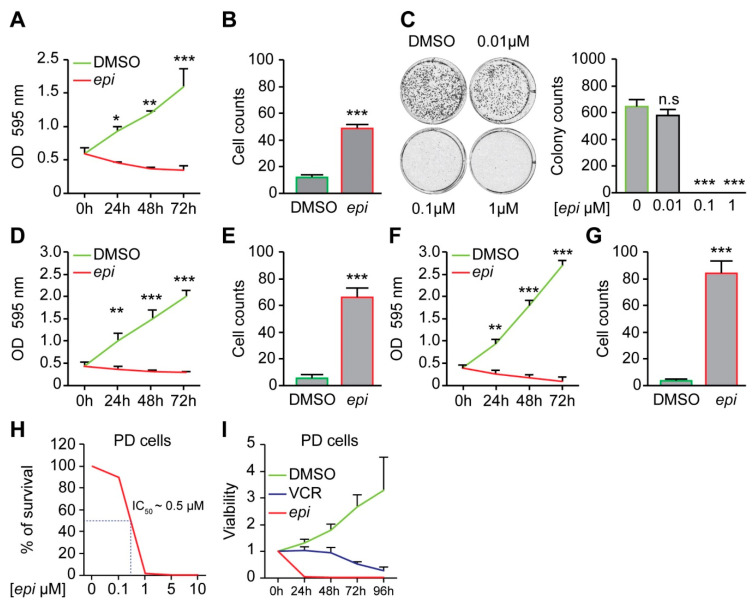
*Epi*-enprioline mediates reduced proliferation and cell survival. (**A**) VCR-20 cells were treated with 0.5 μM *epi*-enprioline and cell proliferation was evaluated at different time points during the course of 72 h. (mean ± SEM; *n* = 3). * *p* < 0.05, ** *p* < 0.01, *** *p* < 0.001 compared to control cells (ANOVA, Tukey’s test). (**B**) VCR-20 cells were treated with 0.5 µM *epi*-enprioline and stained with PI for detecting dead cells (mean ± SEM; *n* = 3). (**C**) VCR-20 cells were treated with the indicated concentrations of *epi*-enprioline for 15 d for clonogenic determinations. Representative well and quantitative data are shown (mean ± SEM; *n* = 3). n.s. = non-significant, *** *p* < 0.001 compared to control cells (ANOVA, Tukey’s test). (**D**) SK-N-AS-VCR-20 cells were treated with 1.2 µM *epi*-enprioline for a crystal violet proliferation assay at different time-points during the course of 72 h. (Mean ± SEM; *n* = 3). (**E**) SK-N-AS-VCR-20 cells were treated with 1.2 µM *epi-*enprioline and stained with PI for detecting dead cells (mean ± SEM; *n* = 3). (**F**) UKF-NB-4-VCR-50 were treated with 3 µM *epi*-enprioline and cell proliferation was evaluated at different time-points during the course of 72 h. (Mean ± SEM; *n* = 3). (**G**) UKF-NB-4-VCR-50 cells were treated with 3 µM *epi*-enprioline and stained with PI for detecting dead cells (mean ± SEM; *n* = 3). (**H**) LU-NB-2 patient-derived xenograft (PDX) cells were treated with *epi*-enprioline and percentage of surviving tumor cells at different concentrations is shown. (*n* = 3). (**I**) Cell viability of treated LU-NB-2 PDX cells was assessed at different time-points using the CellTiter-Glo cell viability assay. (mean ± SEM; *n* = 3). VC = vincristine; epi = epi-enprioline.

**Figure 3 ijms-21-06577-f003:**
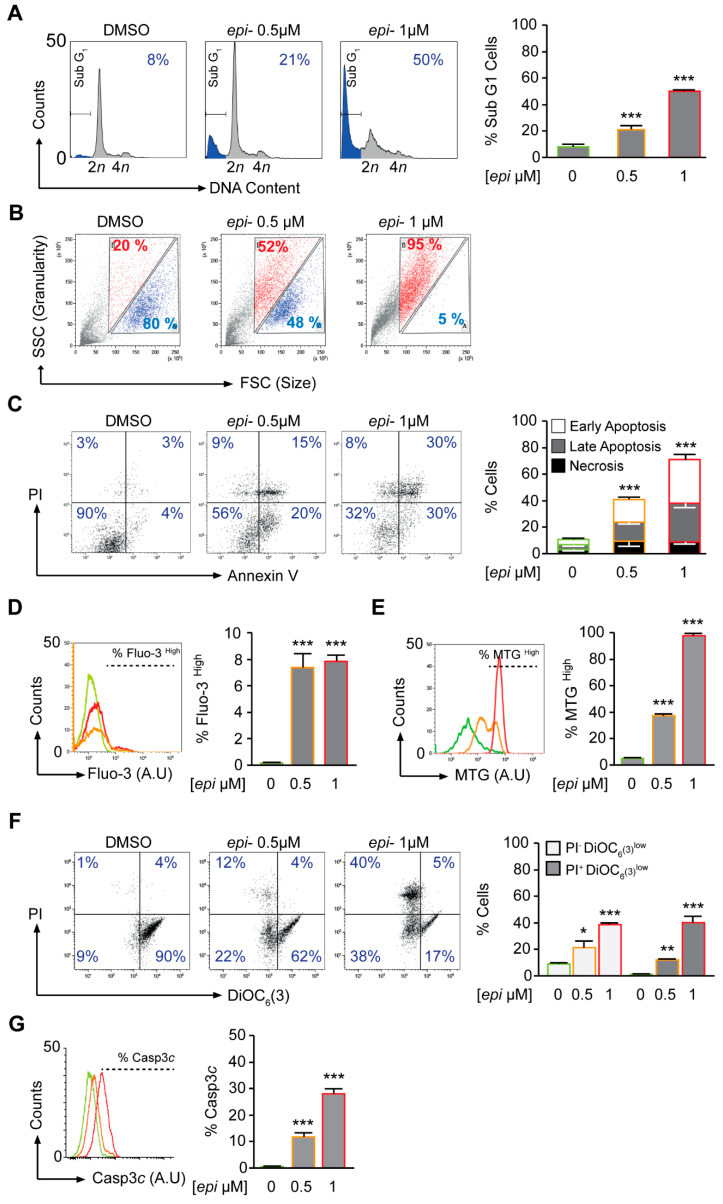
*Epi*-enprioline mediates cell death via the mitochondrial apoptotic pathway. (**A**) Cells were treated with *epi*-enprioline and stained with Hoechst 33342 for the quantification of the subG1 apoptotic population. Representative plots and quantitative data are reported (mean ± SEM; *n* = 3). *** *p* < 0.001 compared to control cells (ANOVA, Tukey’s test). (**B**) VCR-20 cells were seeded and treated with *epi*-enprioline. After 24 h, cells were collected and acquired by flow cytometry. Representative plots are reported with the respective gate (*n* = 3). (**C**) Cells administered with *epi*-enprioline for 24 h were stained for the phosphatidylserine exposure with FITC-conjugated annexin V and the vital dye PI. *** *p* < 0.001 compared to control cells (ANOVA, Tukey’s test, *n* = 4). (**D**) Cells were treated with *epi-*enprioline (labelled in green, orange and red, respectively) and after 24 h, stained with the calcium dye Fluo-3 for quantification of the cytosolic Ca2+ concentration by flow cytometry(mean ± SEM; *n* = 3). *** *p* < 0.001 compared to control cells (ANOVA, Tukey’s test). (**E**) Cells were treated with *epi*-enprioline (labelled in green, orange and red, respectively) and after 24 h, stained with the MitoTracker Green (MTG) dye for the quantification of mitochondrial inner membrane and matrix(mean ± SEM; *n* = 3). *** *p* < 0.001 compared to control cells (ANOVA, Tukey’s test). (**F**) Cells administered with *epi*-enprioline for 24 h were co-stained with the vital dye PI and the mitochondrial membrane potential (Δψm)-sensing dye DiOC6. White columns illustrate early apoptosis (PI−DiOC6 low) while grey columns illustrate late apoptosis (PI+DiOC6 low). * *p* < 0.05, ** *p* < 0.01, *** *p* < 0.001 compared to control cells (ANOVA, Tukey’s test). (**G**) Cells were treated with *epi*-enprioline (labelled in green, orange and red, respectively) and after 24 h labeled with the FITC-conjugated cleaved caspase-3 (mean ± SEM; *n* = 3).

**Figure 4 ijms-21-06577-f004:**
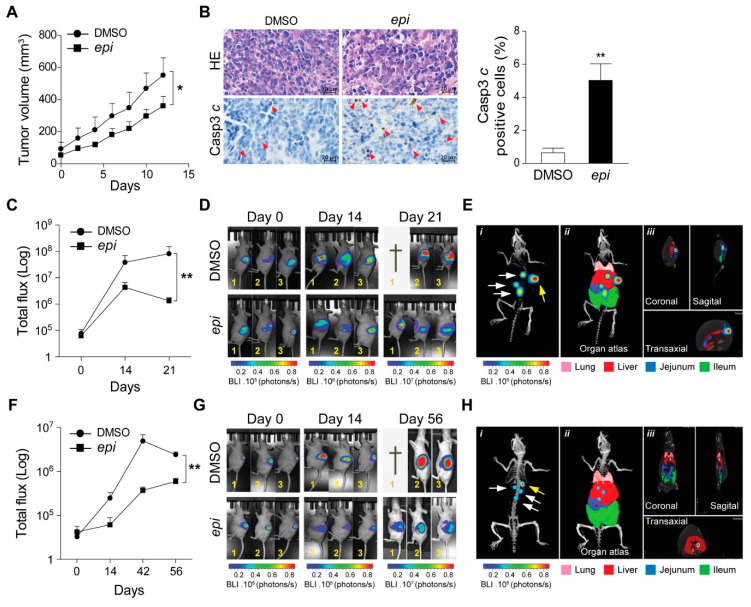
*Epi*-enprioline reduces tumour growth in vivo. (**A**) Nude mice were injected with 2 × 10^6^ VCR-20 cells. When the mice developed visible subcutaneous tumours using caliper (100–125 mm^3^), they were treated intra-tumour, 2–3 cm away from the tumour, with 10 mg/kg of epi-enprioline or DMSO control (*n* = 6 mice in each group). The mean tumor volume for each treatment group was graphed plus/minus standard errors (* *p* < 0.05). (**B**) Immunohistochemistry analysis of tumour biopsies isolated from control (Ctrl) or *epi*-enprioline-treated VCR-20 cells using haemotoxylin and eosin or antibodies against cleaved caspase-3. Arrowheads indicate cleaved caspase-3 positive cells (left panel). Quantification of the number of cleaved caspase-3 (Asp175) positive cells in the sections isolated from control and *epi*-enprioline-treated group (Right panel, mean ± SEM, ** *p* < 0.01, *n* = 3). (**C**) Nude mice were subjected to orthotopic implantation into adrenal gland with VCR-20-GFP-Luc cells (1 × 10^6^) and a week later, the tumor growth was monitored by non-invasive 2D bioluminescence (BLI) imaging using IVIS-CT spectrum. Mice (*n* = 6) showing tumor growth signals were randomized into DMSO and *epi*-enprioline treatment groups. BLI signal intensity was quantified in Total flux (Photons/s) after deducting the average background signal (Bkg) from measurement region of interest (ROI) using the Living image analysis software (mean ± SEM, ** *p* < 0.01). (**D**) Representative images of tumor growth (VCR-20 GFP-Luc cells) in mice treated with DMSO or *epi*-enprioline, † indicates animal not surviving the entire study period. (**E**) Ventral side 3D μCT representative images were taken to show distant metastasis of VCR-20 Luc cells in control animals after orthotopic implantation into adrenal gland. Site of implantation is shown by yellow arrowhead. The mouse organ atlas (ii and iii) was overlaid on the 3D skeletal image (i) to show the possible metastatic anatomical sites (white arrowheads) using Living image analysis software. (**F**) Nude mice were subjected to orthotopic implantation into adrenal gland with LU-NB-2 GFP-Luc (2 × 10^6^). One week later, the tumor growth was monitored by non-invasive 2D bioluminescence (BLI) imaging, using IVIS-CT spectrum. Mice (*n* = 6) showing tumor growth signals were randomized into DMSO or *epi*-enprioline treatment groups. BLI signal intensity was quantified in Total flux (Photons/s) after deducting the average background signal (Bkg) from measurement region of interest (ROI) using the Living image analysis software (mean ± SEM, ** *p* < 0.01). (**G**) Representative images of tumor growth (LU-NB-2 GFP-Luc) in mice treated with DMSO or *epi*-enprioline, † indicates animal not surviving the entire study period. (**H**) Ventral side 3D μCT representative images were taken to show distant metastasis of LU-NB-2 GFP-Luc cells after orthotopic implanted into adrenal gland. Site of implantation is shown by yellow arrowhead. The mouse organ atlas (ii and iii) was overlaid on the 3D skeletal image (i) to show the possible metastatic anatomical sites (white arrowheads) using Living image analysis software.

**Figure 5 ijms-21-06577-f005:**
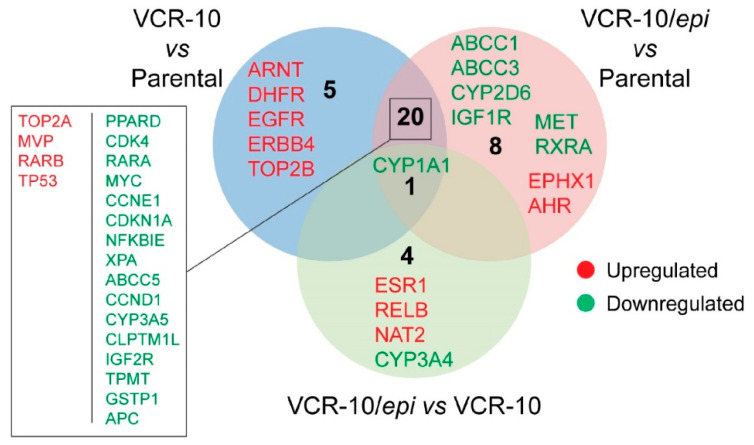
Identification of genes involved in drug resistance in *epi*-enprioline-treated and non-treated VCR-10 cells. The Human Cancer Drug Resistance RT² Profiler PCR Array revealed up- and downregulated genes involved in drug resistance in *epi*-enprioline-treated and non-treated VCR-10 cells. The levels of relative expression of each gene in the two samples are plotted against each other in a Venn-diagram after normalisation was done using the housekeeping genes. The line in the middle indicates relative fold changes. The left and right lines indicate the fold change in gene expression threshold, which was defined as 1.5-fold.

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
