# Peer review of "Discovery of epi-Enprioline as a Novel Drug for the Treatment of Vincristine Resistant Neuroblastoma"

_ijms, 2020, doi:10.3390/ijms21186577_

Round 1
Reviewer 1 Report
Manuscript ID: ijms-896429 review
Comments for the Authors
The manuscript entitled “Identification of epi-enprioline as a novel drug for the treatment of chemoresistant neuroblastoma” by Sime and colleagues reports that Epi-enprioline reduced proliferation and cell survival in NB.
The authors have screened the new anti-cancer drug for vincristine-resistant NB cells and have found that epi-enprioline induced cell apoptosis via mitochondrial depolarization.
Paper is well organized and author screening also quite good.
However, in its present form the manuscript raises several questions. The authors should explain and address these questions.
Critical major issues to be addressed:
- Author showed western blotting data (Fig. 2A) and suggested epi-enprioline doesn’t affect cytoskeleton structure alteration. However, WB data cannot prove cytoskeleton structure alteration. Author should showed cytoskeleton structure using IF.
- Does epi-enprioline also affect normal cells? Author should include normal cell and show cell viability.
- For several experiments, it is unclear how many times the experiment was repeated.
Minor corrections:
- Author should move Figure 2B to 4B.
- Author should use constant cell name in figure and manuscript (Fig. 5).
- In Figure 4B, author showed as PI+, Annexin V+ population is only late apoptosis. However, this population also included necrosis population. Author should correct graph.
Author Response
Response to Reviewer #1
Comments for the Authors
The manuscript entitled “Identification of epi-enprioline as a novel drug for the treatment of chemoresistant neuroblastoma” by Sime and colleagues reports that Epi-enprioline reduced proliferation and cell survival in NB. The authors have screened the new anti-cancer drug for vincristine-resistant NB cells and have found that epi-enprioline induced cell apoptosis via mitochondrial depolarization.
Paper is well organized and author screening also quite good.
However, in its present form the manuscript raises several questions. The authors should explain and address these questions.
We thank the Reviewer for the careful review of our manuscript and the valuable and constructive suggestions.
Critical major issues to be addressed:
Author showed western blotting data (Fig. 2A) and suggested epi-enprioline doesn’t affect cytoskeleton structure alteration. However, WB data cannot prove cytoskeleton structure alteration. Author should showed cytoskeleton structure using IF.
Response:
The reviewer is correct that WB data cannot prove cytoskeleton structure alteration. We have preliminary data showing that action filaments or microtubules structures were not altered in epi-enprioline-treated cells. However, based on the comment from the reviewer #2 suggesting to “simplify the paper” and “describe only those experiments which really help to understand what is going on around epi-enprioline” we decided to removing any data concerning the possible mechanism of epi-enprioline including the WB in Fig. 2. We will investigate the mechanism of action of epi-enprioline in more details in future studies.
Does epi-enprioline also affect normal cells? Author should include normal cell and show cell viability.
Response:
To address this question, we have used primary cells isolated from mouse embryonic fibroblast and it appeared that these cells were not affected by epi-enprioline-treatment suggesting that epi-enprioline targets chemoresistant cancer cells. This new data is present as supplementary Fig. 2. Include n in the fig legends Simplify the text from the legend
For several experiments, it is unclear how many times the experiment was repeated.
Response:
We have carefully looked through the manuscript and included “n” for the missing experiments in the figure legends.
Minor corrections:
Author should move Figure 2B to 4B.
Response:
We have moved figure 2B to 4B.
Author should use constant cell name in figure and manuscript (Fig. 5).
Response:
In the revised version of our manuscript we are using only the name “VCR-10” or “VCR-20” in the figure and in the text.
In Figure 4B, author showed as PI+, Annexin V+ population is only late apoptosis. However, this population also included necrosis population. Author should correct graph.
Response:
To set up the demarcation of the cell death, we are using the parameters (PI+ and Annexin V-) for necrotic cells, and (PI+, Annexin V+) for late apoptotic cell population in both DMSO and epi-treated cells. The percentage of these two populations has been shown in figure 4B (now Fig. 4C).
Reviewer 2 Report
This is excellent paper which describes finding with a new possible drug against neuroblastoma. The conclusion is highly underlined by the described experiments and their results. However, this is a rather busy paper. There are several cell lines used with different properties Nmyc amplified/not amplified, VCR resistant in different rate, primary tumor, metastasis, relapse, etc. The reader finds too much results, and a little bit hard to follow which experiment why really happened, and which cell lines was used to prove what kind of statement. I suggest to simplify the paper, describe only those experiments which really help to understand what is going on around epi-enprioline in neuroblastoma and authors should hold reader's hand stronger during epi-enprioline journey. Neuroblastoma treatment is not failed only due to VCR resistance, as several other more potential drug also part of its treatment. There should be a clear explanation why authors found highly important to approach neuroblastoma chemotherapy resistance through VCR resistance. According to Fig 6 I PD cells are sensitive to VCR, and in higher rate to epi-enprioline, too. What this experiment would like to prove? I suggest to make clearer the presumed role of epi-enprioline against neuroblastoma. This is generally a new drug against neuroblastoma, which could be used in all cases or this is especially important in case of VCR resistance? Depending of the answer of this question I would rewrite the title and all paper, focusing on the final conclusion. Minor comment: 1. "NB" as an abbreviation is never solved, 2. I suggest to use one name for each cell type consequently in the whole paper: e.g. "Twenty ng/ml VCR-resistant cells" vs "SK-N-ASrVCR20" vs "VCRresistant (10 and 20) neuroblastoma cells" to make it easier to follow 3. Figure legends should be more concise. Description of the experiment should not described there, but short interpretation of the findings would help a lot to understand the Figures.Author Response
Response to Reviewer #2
This is excellent paper which describes finding with a new possible drug against neuroblastoma. The conclusion is highly underlined by the described experiments and their results. However, this is a rather busy paper. There are several cell lines used with different properties Nmyc amplified/not amplified, VCR resistant in different rate, primary tumor, metastasis, relapse, etc. The reader finds too much results, and a little bit hard to follow which experiment why really happened, and which cell lines was used to prove what kind of statement. I suggest to simplify the paper, describe only those experiments which really help to understand what is going on around epi-enprioline in neuroblastoma and authors should hold reader's hand stronger during epi-enprioline journey. Neuroblastoma treatment is not failed only due to VCR resistance, as several other more potential drug also part of its treatment. There should be a clear explanation why authors found highly important to approach neuroblastoma chemotherapy resistance through VCR resistance. According to Fig 6 I PD cells are sensitive to VCR, and in higher rate to epi-enprioline, too. What this experiment would like to prove? I suggest to make clearer the presumed role of epi-enprioline against neuroblastoma. This is generally a new drug against neuroblastoma, which could be used in all cases or this is especially important in case of VCR resistance? Depending of the answer of this question I would rewrite the title and all paper, focusing on the final conclusion.
We thank the Reviewer for the careful review of our manuscript and the valuable and constructive suggestions.
However, this is a rather busy paper. There are several cell lines used with different properties Nmyc amplified/not amplified, VCR resistant in different rate, primary tumor, metastasis, relapse, etc. The reader finds too much results, and a little bit hard to follow which experiment why really happened, and which cell lines was used to prove what kind of statement. I suggest to simplify the paper, describe only those experiments which really help to understand what is going on around epi-enprioline in neuroblastoma and authors should hold reader's hand stronger during epi-enprioline journey.
Response:
We have simplified the manuscript by removing and rearranging figures. In the revised version the story starts and end according to the following:
Fig. 1: Identification of epi-enprioline
Fig. 2: The effect of epi-enprioline in cell survival and cell proliferation
Fig. 3: Epi-enprioline-mediates cell death via intrinsic mitochondrial apoptotic pathway
Fig. 4: In vivo effect of Epi-enprioline
Fig. 5: Alteration in selected gene expression and signaling pathways in epi-enprioline treated cells
Neuroblastoma treatment is not failed only due to VCR resistance, as several other more potential drug also part of its treatment. There should be a clear explanation why authors found highly important to approach neuroblastoma chemotherapy resistance through VCR resistance.
Response:
The arguments for using Vincristine resistant cells was based on the i) importance of usage of this drug in other neuronal cancer diseases such as treatment of advanced bilateral intraocular retinoblastoma and children with newly diagnosed progressive low-grade gliomas. Secondly, in these diseases, similar to NB, the resistance is commonly occurring and ii) the mechanism behind it is not known. Furthermore, iii) drug resistance to Vincristine in contrast to other commonly used drugs is mediated by both over expression and amplification of the certain genes. These are the major reasons why our study was focused on Vincristine resistant cells.
According to Fig 6 I PD cells are sensitive to VCR, and in higher rate to epi-enprioline, too. What this experiment would like to prove? I suggest to make clearer the presumed role of epi-enprioline against neuroblastoma. This is generally a new drug against neuroblastoma, which could be used in all cases or this is especially important in case of VCR resistance? Depending of the answer of this question I would rewrite the title and all paper, focusing on the final conclusion.
Response:
The experiment in Fig. 2I (now 3I) shows that epi-enprioline is much more effective compared to Vincristine in chemo-resistant patient derived cells. Since we do not present any additional data investigating whether epi-enprioline can be more effective in other chemo-resistant cells beside Vincristine, we have decided to replace “chemo-resistant” with “vincristine-resistant” throughout the manuscript including the title. In the discussion we have also incorporated the following sentence: “Further studies need to establish whether the anti-cancer function of epi-enprioline is restricted to NB or whether it also applies to other chemo reagents and chemoresistant human cancer cells, in addition to future drug toxicology studies” (page 15).
Minor comment: 1. "NB" as an abbreviation is never solved,
Response:
We have inserted the full name for NB, page 1 line 33 and page 4 line 163.
- I suggest to use one name for each cell type consequently in the whole paper: e.g. "Twenty ng/ml VCR-resistant cells" vs "SK-N-ASrVCR20" vs "VCRresistant (10 and 20) neuroblastoma cells" to make it easier to follow
Response:
We have changed and use only the name VCR-10 or VCR-20 throughout the manuscript.
- Figure legends should be more concise. Description of the experiment should not described there, but short interpretation of the findings would help a lot to understand the Figures.
Response:
We have shortened all the figure legends in the manuscript.
Round 2
Reviewer 1 Report
In the revised manuscript by Dr. Massoumi and his/her colleagues, the authors answered almost all of my concerns. Thus, since the substantial improvement, the manuscript should be considered for publication.
Author Response
We thank the Reviewer for the careful review of our manuscript and the valuable and constructive suggestions.
Reviewer 2 Report
The paper highly improved.
The mass around nomeclature of cell types is still not completely solved.
Minor corrections are needed:
1. Now, authors use "VCR-20" , "VCR-10" correctly, throughout the paper, but these cell types are still not clearly defined in Materials and Methods section.
2. Fig 4A-B, Number of mice is not defined of each group in the text or in Figure label
Author Response
August 31, 2020
Response to Reviewer
Comments and Suggestions for Authors
The mass around nomeclature of cell types is still not completely solved.
Response:
We thank the Reviewer for suggestions.
All cell lines used in the study is now defined in Material and Methods.
These cells include:
Be2c-parental cells:
The human NB cell line SK-N-BE(2)-C (ATCC, CRL-2268, parental) was cultured at 37°C and 5% CO2 in minimum essential medium (MEM, HyClone, Thermo Fisher Scientific, MA, USA) supplemented with 10% fetal bovine serum (FBS) (Sigma-Aldrich Sweden AB, Stockholm, Sweden) and 1% penicillin/streptomycin (GIBCO, MA, USA).
VCR-10 and VCR 20:
To generate vincristine-resistant cells (Be2c-VCR), SK-N-BE(2)-C cells were cultured for 7 months with gradually increased concentrations of vincristine ranging from 1-10 ng/ml. The 10 ng/ml Be2c-VCR cells (VCR-10) were treated further with escalating concentration of 20 ng/ml (VCR-20) of vincristine for another month.
SK-N-AS and UKF-NB-4:
The non-MYCN-amplified neuroblastoma cell line SK-N-AS was kindly provided by Dr. Angelika Eggert (Universität Duisburg-Essen, Germany). The MYCN-amplified, TP53-mutant (C175F) neuroblastoma cell line UKF-NB-4 was derived from bone marrow metastases of a patient harboring a stage IV neuroblastoma (18, 19).
SK-N-AS-20 and UKF-NB-4-50
The vincristine-resistant sub-lines were established by continuous exposure to step-wise increasing drug concentrations as previously described (20) and derived from the Resistant Cancer Cell Line (RCCL) collection (https://www.kent.ac.uk/stms/cmp/RCCL/RCCLabout.html). UKF-NB-4 cells were adapted to growth in the presence of vincristine 50 ng/mL (UKF-NB-4-VCR-50), SK-N-AS to vincristine 20 ng/mL (SK-N-AS-VCR-20). Cells were grown at 37 °C in Iscove’s modified Dulbecco’s medium supplemented with 10% heat-inactivated foetal calf serum and containing 100 IU/ml of penicillin and 100 μg/ml streptomycin. The vincristine-adapted sub-lines were continuously cultivated in the presence of the indicated drug concentrations.
LU-NB-2:
LU-NB-2 patient-derived xenograft (PDX)-cells were established and characterized as previously described (21, 22). Briefly, PDX cells were derived from a NB brain metastasis following treatment relapse. Tumor cells contained 1p del, MYCN amplification and 17q gain. PDX cells were cultured as free-floating 3D-spheres in serum-free medium consisting of Dulbecco’s Modified Eagle’s Medium (DMEM) and GlutaMAXTM F-12 (3:1 ratio) supplemented with 1% penicillin/streptomycin, 2% B27 w/o vitamin A, 40 ng/µl basic Fibroblast Growth Factor and 20 ng/µl Epidermal Growth Factor.
Minor corrections are needed:
1. Now, authors use "VCR-20" , "VCR-10" correctly, throughout the paper, but these cell types are still not clearly defined in Materials and Methods section.
Response:
We have included the definition for these cells in Materials and Methods under “Cell culture” (page 2)
The 10 ng/ml Be2c-VCR cells (VCR-10) were treated further with escalating concentration of 20 ng/ml (VCR-20) of vincristine for another month.
2. Fig 4A-B, Number of mice is not defined of each group in the text or in Figure label.
Response:
The number of the mice were indicated in the Materials and Methods under “Animal model”:
“VCR-20 cells (2.0 × 106 cells in 0.1 mL phosphate buffered saline (PBS)) were subcutaneously injected into the right flank of 4-week-old female NMR1-Foxb1 nude mice (Janvier Labs, France). Mice were randomly assigned into two weight-matched groups with 6 mice in each group receiving either epi-enprioline or DMSO”
We have included the number of mice in the figure legend for Fig. 4A and 4B.